# Bio-Inspired Strategies Are Adaptable to Sensors Manufactured on the Moon

**DOI:** 10.3390/biomimetics9080496

**Published:** 2024-08-15

**Authors:** Alex Ellery

**Affiliations:** Centre for Self-Replication Research (CESER), Department of Mechanical & Aerospace Engineering, Carleton University, 1125 Colonel By Drive, Ottawa, ON K1S 5B6, Canada; aellery@mae.carleton.ca; Tel.: +1-613-261-3765

**Keywords:** sensors, biomimetic sensing, biomimetic vision, in situ resource utilization, lunar resources, lunar industrialization

## Abstract

Bio-inspired strategies for robotic sensing are essential for in situ manufactured sensors on the Moon. Sensors are one crucial component of robots that should be manufactured from lunar resources to industrialize the Moon at low cost. We are concerned with two classes of sensor: (a) position sensors and derivatives thereof are the most elementary of measurements; and (b) light sensing arrays provide for distance measurement within the visible waveband. Terrestrial approaches to sensor design cannot be accommodated within the severe limitations imposed by the material resources and expected manufacturing competences on the Moon. Displacement and strain sensors may be constructed as potentiometers with aluminium extracted from anorthite. Anorthite is also a source of silica from which quartz may be manufactured. Thus, piezoelectric sensors may be constructed. Silicone plastic (siloxane) is an elastomer that may be derived from lunar volatiles. This offers the prospect for tactile sensing arrays. All components of photomultiplier tubes may be constructed from lunar resources. However, the spatial resolution of photomultiplier tubes is limited so only modest array sizes can be constructed. This requires us to exploit biomimetic strategies: (i) optical flow provides the visual navigation competences of insects implemented through modest circuitry, and (ii) foveated vision trades the visual resolution deficiencies with higher resolution of pan-tilt motors enabled by micro-stepping. Thus, basic sensors may be manufactured from lunar resources. They are elementary components of robotic machines that are crucial for constructing a sustainable lunar infrastructure. Constraints imposed by the Moon may be compensated for using biomimetic strategies which are adaptable to non-Earth environments.

## 1. Introduction

Much of biomimetics is focussed on emulating biological materials which often combine polymeric elasticity with ceramic hardness [1]. However, in a hostile world, biological organisms exhibit autonomy, adaptability, robustness, lightweight construction and self-repair. These are highly desirable characteristics for robotic systems. There have been many applications of biomimetics to space missions despite the space environment being different to that driving biological evolution on Earth, e.g., gecko foot dry adhesion, spider silk, jumping spider actuation, insect campaniform sensilla, woodwasp ovipositor, etc. [2,3]. There are numerous opportunities to exploit biomimetic solutions evolved on Earth to spacecraft off-Earth.

The key lesson is that biomimetics imparts the adaptability and robustness of miniaturized biological organisms to engineering systems, in this case, spacecraft. An example of biomimetic solutions for space debris removal includes hair-based tactile sensing, gecko foot adhesion, bee stinger harpoon, animal jaws, venus flytrap, octopus grappling, woodpecker shock absorption, flower folding of drag sails and swarm behavior [4]. We investigate the implications of biomimetics for advanced in situ resource utilization (ISRU) devoted to lunar industrialization. We shall find that biomimetics offers solutions to limitations imposed by locally-manufactured robotics required for full lunar industrialization. It is remarkable that biological solutions evolved to solve Earth-encountered problems can be applied to the development of human-created technologies for deployment onto the Moon, a sterile world inimicable to life and with little in common with its neighbor Earth.

The key to full lunar industrialization is the exploitation of lunar resources for the manufacture of the robotic machines that build infrastructure (Figure 1). The cost of launching large-scale assets from the Earth and landing them on the Moon is prohibitive. This is true for large-scale robotic machines to build infrastructure on the Moon so such robotic machines must be manufactured in situ from lunar resources. We assume that the dominant form of manufacture on the Moon will be 3D printing (additive manufacturing). The advantage of additive manufacturing over subtractive manufacturing methods (such as milling) is that it requires no specialized tooling, produces little or no waste and is highly versatile in the complexity of its printed structures. For example, 3D printing has been proposed for building lunar bases by contour crafting [5] or D-shaping [6].

On Earth, 3D printing has been applied partially to manufacture robots whereby polymer deposition and machining has manufactured compliant joints with cavities into which pre-existing actuator and sensing components may be embedded during manufacture [7]. An extension of this is the use of polyjet 3D printing of a biomimetic finger with viscoelastic tendons within rigid plastic phalanges driven by artificial muscles of pneumatic bellows [8]. Today, laser-based additive manufacturing offers biomimetic design of metals and polymers to fabricate biomimetic structures such as butterfly wing, webbing, honeycomb and tensegrity-type structures [9]. On the Moon, prior to 3D printing of parts and components, there is a sequence of processes that must be undertaken to utilize in situ resources from their native form. The lunar industrial architecture comprises a sequence of manufacturing processes all supported by solar power generation/flywheel energy storage stations [10], rovers for surveying and excavating lunar regolith [11,12], an electromagnetic/electrostatic separation station for beneficiating regolith, a unit chemical reactor for acid leaching to extract pure oxides [13], an electrochemical reactor for reducing pure oxides into mineral [14,15] and a 3D printing station for 3D printing components [16] that are subsequently assembled using the same 3D printing station as a cartesian assembly robot. Such an industrial capacity can support construction of lunar bases [17] and their life support systems [18].

The first biomimetic strategy—similar to that faced by the emergence of life on Earth—is to function within the material availability in the environment. So it must be with an artificial robotic machine on the Moon [19]. Any hierarchical robotic system is founded on a servo-level control system comprising actuators, sensors and feedback controller upon which more sophisticated functions may be configured either through design (most commonly) or emergence (such as in evolutionary robotics). One example of such a robotic system would be a self-replicating machine which must be premised on its fundamental components [20]. In both cases, fundamental functional components are the foundation of any control hierarchy, be it biological or robotic. This will be our focus here—on foundational functions including information processing from which hierarchies emerge. TRIZ (Teorija Reshenija Izobretatel’skih Zadach) analysis suggests that engineering and biological solutions to similar problems are 88% divergent and that technological solution exploit energy at the expense of mechanism and information that are utilized by biology [21]. Here, we explore structural recipes for the technological construction of components that perform information processing of environmental signals.

Machines of production, namely mining and manufacturing machines, are robotic machines necessary to construct lunar infrastructure, which is itself leveraged from lunar resources. Our focus is on more fundamental components of robotic machines. Robotic machines of production include rovers to excavate lunar regolith, ball mills to comminute regolith, electrostatic separators to beneficiate regolith, pumps to drive unit (electro)chemical reactors, 3D printers and other supplementary manufacturing machines to fabricate parts and assembly manipulators to construct systems. All these robotic machines involve different kinematic configurations of electric motors with their attendant control systems and sensors. The elementary feature of all robot machines is their control system—control electronics map sensory data to motor-driven behavior. Feedback is fundamentally premised on sensors that provide measurements of environmental properties. Sensors represent the most sophisticated and challenging components to manufacture in situ. As we focus on material resources availability on the Moon, our investigation includes feasible chemical processing pathways on the Moon. We examine this for each sensor but this resides within the context of a broader chemical processing architecture for general lunar resource processing—the lunar industrial ecology [22] (Appendix A) which outlines a suite of chemical processes that are linked through an ecology in which the waste of one process provides feedstock for another in a system of interlocking recycling loops. To overcome some of the deficiencies imposed by lunar resources, we apply bio-inspired principles to the control systems of such sensors to compensate. It is worth noting at the outset that given the difficulties in solid-state technology manufacture on the Moon [23], we shall not be considering modern solid-state sensor microtechnology.

Dimensional analysis provides the starting point for the measurement of physical properties. The fundamental dimensions are—amount, mass, length, time, temperature, electric current and luminosity. All sensory transducers are derived from these elemental parameters. Our concern here is the in situ construction of sensors from lunar resources, although in foveated vision, sensor and actuator functions are entwined. As we shall see, a bio-inspired approach to the control of such sensors is indispensable. Biomimetics is particularly applicable to the constraints of sensors manufactured on the Moon from lunar resources. We are concerned with touch (including proprioception) and vision sensing as the most relevant sensors for robotic applications. Proprioception may be regarded as a derivative of tactility that activates the somatosensory system, which is itself comprised of two subsystems. The cutaneous subsystem processes tactile data from the skin. The kinaesthetic subsystem processes proprioceptive data from the muscles. Together, they provide haptic information that support social interactions [24] but it is the sensorimotor functions that are important here. The design of the sensors themselves is not bio-inspired but is premised on conventional technology adapted to the lunar environment. However, the constraints imposed by lunar resources impose performance limitations on such designs.

We focus on two classes of sensor—position sensors and derivatives thereof as foundational measurements and light sensing arrays for general remote distance measurement. These two sensor modalities measure internal state (displacement and force) and external state (visual reflectance) respectively which are crucial for feedback control systems in robotic devices—imaging cameras and displacement/tactile sensing of actuation effects. We first consider the hardware requirements of this minimal sensor suite to include displacement sensors, tactile sensors and vision sensors constructed within the constraints of lunar resources. First, we examine electrical resistance of metals and piezoelectricity of quartz as transduction mechanisms for measuring basic mechanical parameters. Thence, the photomultiplier tube (PMT) is highlighted as the pixel element for vision and is a sophisticated showcase sensor that could be manufactured from lunar resources. Lunar resource availability imposes performance limits on our sensors but bio-inspired approaches can compensate for these limitations. The coarse resolution from arraying PMTs imposes severe performance limits. We then look to biological vision to determine if there are lessons we can learn to overcome this major deficiency. First, we can adopt a divide-and-conquer strategy to separate object identification from object location. For object identification, foveated vision offers the prospect for sub-pixel imaging. For object location, optical flow vision offers the prospect for visual navigation.

This paper presents a roadmap to manufacturing sensors on the Moon emphasizing lunar materials and aspects of their chemical processing rather than manufacturing techniques which are of course fundamentally premised on the former. The Moon represents an inhospitable environment that introduces significant further complexities. It is a high vacuum, high radiation and, in particular, low-gravity environment which will have major impacts on manufacturing. The high vacuum environment can be exploited such as in electron beam processing techniques requiring a vacuum. The high radiation environment degrades hydrocarbon plastics (but less so silicone plastics) and has detrimental effects on solid-state electronic devices (but less so on vacuum tube devices). The low-gravity environment of one-sixth g will have effects on manufacturing processes where gravity is important but the degree of severity is currently unexplored—we list a few manufacturing processes that will be affected: (i) anchorage through reaction weight for excavating and drilling; (ii) electrostatic and other separation techniques; (iii) fluids, both liquids and gases, will experience reduced buoyancy-driven convection while Marangoni convection effects will become more pronounced; (vi) there will be increased tendency to delamination during 3D printing processes. Clearly, these are significant factors that must be addressed in actual situ manufacturing but we do not do so here.

## 2. Displacement and Cognate Parameters Sensing

Position is the most basic measurement from which many other mechanical parameters are derived. The simplest position sensor is the potentiometer which is a variable resistance wire in a voltage divider configuration. The resistance wire may be made from aluminium which is extractable from lunar anorthite, e.g., [13,15,25]. Temperature, strain, stress, pressure, acceleration and force measurements may be derived from resistance measurement. Strain, temperature and relative humidity sensors have been 3D printed by stereolithography, polymer extrusion, laminated object manufacturing, inkjet and screen-printing of conductive metals (Ag or Al), conductive polymers (PEDOT:PSS (poly(3,4-ethylenedioxyrgophene) polystyrene sulphonate)) and piezoelectric polymers (PVDF (poly(vinylidenefluoride))) on rigid or flexible substrates [26,27]. Hydrocarbon polymers, however, are impractical on the Moon due to the limited availability of elemental or compound forms of carbon. Silicone elastomers and oils such as PDMS may be synthesized in situ on the Moon from the limited carbon volatiles implanted in lunar regolith and silicate minerals. Carbon volatiles are the most common volatiles (except for water) at ~0.01% by mass while water ice for hydrogen feedstock exists in regolith in polar regions at 5–6% by mass. Heating regolith releases water vapour at −53 °C under hard vacuum and carbon and other adsorbed volatiles at >600 °C. Silicon is sourced from lunar silicate minerals, e.g., HCl leaching of anorthite yields silica which is reduced to silicon through molten salt electrolysis [13,15]. From these volatile resources, CO_2_ and H_2_ feedstock is converted into siloxanes such as PDMS through the Rochow process (see Appendix A). However, silicone polymer with its alternating Si-O backbone requires C only for its side chains, thereby minimizing C consumption compared with hydrocarbon plastics.

Strain sensors may be capacitative, piezoresistive or piezoelectric. A capacitative force sensor may be 3D printed with extruded dielectric polymer such as elastomeric polydimethylsiloxane (PDMS silicone rubber) or ceramic ink followed by metal foil lamination to form the plates. Capacitative sensors have good sensitivity and large dynamic range but are susceptible to noise. Metal may also be screen printed or inkjet printed as strain gauges. Strain gauges are meandering metal strips that exhibit a change in electrical resistance when deformed under stress. Suitable metals include Al and Ni, which are derivable from lunar resources: (i) HCl acid leaching of lunar anorthite yields alumina [13] which may be electrolytically reduced to Al metal [15]; (ii) Ni is a major constituent of M-type asteroid material that may be buried in or delivered to the Moon [28]. A Wheatstone bridge circuit measures the resistance change in the strain gauge.

Behavioral response to tactile stimuli in multicellular organisms pre-date the evolution of neurons [29]—it exists in primitive multicellular animals such as sponges and in single-celled ciliates. In biological organisms, flagellar-mediated bacterial chemotaxis taxis implicate touch sensing in bacteria with computation implemented by chemical amplifiers and switches of metabolic reactions [30]. The insect campaniform sensillum and spider slit sensillum operate similarly as a strain gauge, comprising an elliptical hole within a cuticle plate with fibers of chitin surrounding the hole [31]. Tactility is crucial to any agent’s sensing capability, biological or artificial.

Pressure, defined as (force/unit area), is the foundation of tactile sensing (taction). Tactile sensors detect deformation generated by pressure imposed by physical contact with the environment. Tactile sensing is a sophisticated sensory mode of the skin with a spatial resolution of ~1 mm^2^ mediated by four mechanoreceptors: (a) Pacinian corpuscles reside deep in the dermis with a fast response to both vibration and touch; (b) Meissner corpuscles reside just under the epidermis with a moderate response to both touch and rate of touch; (c) Merkel discs have slow response to touch; and (d) Ruffini endings have slow response to pressure and temperature. Slowly adaptive (SA) response (Merkel discs and Ruffini cylinders) is emulated through piezoresistive sensing of rough texture and fast adaptive (FA) response (Meissner corpuscles and Pacinian corpuscles) through piezoelectric sensing of vibrations. Furthermore, Meissner cells respond to low-frequency vibrations while Pacinian cells respond to high-frequency vibrations [32]. Coarse texture with roughness in excess of 200 µm generates a moderate response (with a spectral resolution 50 Hz) of the Meissner receptors. Fine texture with roughness below 200 µm generates a fast response (with a spectral resolution of 250 Hz) of the Pacinian receptors. In proprioception, feedback from muscles originates from the spindles (position-derivative data) and Golgi tendon sensors (force data) which provide local force feedback.

Hair sensors are ubiquitous in biology for sensing touch, vibration, auditory, inertial, fluid flow, pressure, temperature and chemical sensing. Biological hairs are embedded in filiform and campaniform sensilla to measure mechanical stress [33]. Filiform sensilla are elliptically shaped hairs that are square root cones whose diameter is proportional to the square root of the distance from the tip [34]. The rigid hair is an inverted pendulum anchored to a spring, sensitive to fluid flow (though hairs are not actually rigid [35]). Campaniform sensilla are unique to insects and measure exoskeleton strain. They comprise an oval dome cuticle ringed by a thickened cuticle. Analogues of strain sensilla include piezoelectric or capacitive transduction [36]. Biomimetic tactile signal processing of spike trains from biomimetic tactile sensors permits rate coding and/or spike time coding using Izhikevich neurons [37]. An artificial cilia bundle comprising an array of PDMS micro-pillars of graded heights was connected by PVDF piezoelectric tip links and encased in hydrogel [38]. Manufactured by lithography [39], they measured flow velocity and flow direction through the tip links by hair deflection rather than substrate strain. Unfortunately, lithography cannot be conducted on the Moon [23], which eliminates the option of cilia.

Larger whiskers comprise arrays of flexible elastomers mounted onto pressure/force-sensitive load cells—they implement indirect transduction for tactile sensing [40]. Each whisker deflects due to linear acceleration, gyroscopic rotation, fluid flow, etc. At the root, the reaction force and moment is given by:(1)$$F=\frac{3\pi Ed^{4}∆}{64L^{3}}\;and\;N=\frac{3\pi Ed^{4}∆}{64L^{2}}$$
where E = Young’s modulus of whisker, d = diameter of whisker, Δ = tip displacement, and L = length of whisker. High force F and moment N measured at the base of a whisker favours short whisker length L but this limits its operational length. There is particularly high sensitivity of base forces F and moments N to increased whisker diameter d. Generally, reduced bending is favoured through high stiffness E, higher whisker thickness and reduced whisker length. Whiskers represent an universal biological sensor modality for measuring a wide variety of mechanical effects that have useful engineering applications. Flies have dual sets of wings, the hind wings of which comprise halteres [41] which are, in essence, high stiffness, short length and thick diameter whiskers. The halteres oscillate vertically at 150 Hz antiphase with respect to the front wings. Each haltere is embedded in 400–500 strain sensors that detect Coriolis forces imposed by changes in orientation. A bio-inspired gyroscope based on blowfly halteres was an engineered analogue [42].

Direct tactile sensing uses different transduction mechanisms but they have many common features. The simplest tactile sensor involves a layer of silicone rubber that overlies an embedded force sensor array. Force and pressure sensing may be implemented as intersections of an orthogonal network of aluminium wiring (sourced from lunar anorthite). A meandering strain gauge sandwiched between two prestrained silicone rubber substrates [43] is a variation of this concept. Capacitive sensing of deformation in silicone rubber such as PDMS is given by:(2)$$F=\frac{CV^{2}}{2d}$$
where $C=\frac{\varepsilon_{0}\varepsilon_{r}A}{d}$ = capacitance and V = voltage. Force sensitivity F is enhanced primarily by a narrow distance d between capacitor plates. Similarly, a change in electrical resistance in an aluminium conductor generates a piezoresistive effect due to an applied force changing the conductor’s dimensions:(3)$$\frac{\Delta R}{R}=\frac{\Delta L}{L}\left(1+2\nu \right)+\frac{\Delta \rho }{\rho }$$
where R = electrical resistance, L = length, ν = Poisson’s ratio and ρ = resistivity. The change in resistance ΔR is slightly more sensitive to the change in length ΔL than to the change in electrical resistivity due to the piezoresistive effect. Embedding the piezoresistive element in elastomeric PDMS increases the sensitivity. PDMS may be enhanced with embedded graphite particles for piezoresistivity. Polysiloxanes (RSiO_1_._5_) may be converted into piezoresistive SiOC ceramics by pyrolysis in an inert atmosphere at or above 1400 °C giving a high piezoresistive sensitivity of ~145 [44]. The piezoresistivity of carbon black embedded within PDMS gives decreasing electric resistance with increasing applied pressure [45]. The composite resistance is given by:(4)$$R=\left(\frac{L}{N}\right)\left(\frac{8\pi hs}{3A^{2}\gamma e^{2}}\right)e^{\gamma s}$$
where h = Planck’s constant, L = number of carbon particles through a single conducting path, N = number of conducting paths, s = silicone insulation thickness between conducting particles, A = effective cross-sectional area, e = electronic charge, $\gamma =\frac{4\pi }{h}\sqrt{2m\varphi }$, m = electron mass, and φ = potential barrier height. In terms of sensor design, the electrical resistance R is sensitive to the density of carbon particles (L/N) and the silicone insulation thickness s. The change in resistance due to applied stress σ with respect to a reference resistance R(0) is given by:(5)$$\frac{R(\sigma )}{R(0)}=\left(1-\frac{\sigma }{M}\right)exp\left\{-\frac{4\pi \sqrt{2m\varphi }}{h}D\left[{\left(\frac{\pi }{6\varphi }\right)}^{1/3}-1\right]\frac{\sigma }{M}\right\}$$
where M = silicone compressive modulus, D = diameter of carbon black particles, and φ = particle volume fraction. The higher the applied stress σ, the lower the resistance R(σ) with respect to the reference resistance R(0). Viscoelastic ink may be extruded into a liquid silicone elastomer directly to fabricate strain sensors [46]. Carbon black particles suspended in silicone oil comprises the ink that forms a resistance network within the liquid silicone matrix (ecoflex).

Silicone rubber may be infused with microfluidic channels filled with conductive fluid. Any applied strain will alter the fluid’s electrical resistance. There are several options for the conductive fluid: (a) conductive solids in the form of fibers or particles, e.g., iron nanoparticles or carbon black suspended in silicone oil; (b) Ga-In alloy may form electrical circuitry to sense pressure or strain in artificial skin [47]; (c) electrorheological (ER) with conducting (aluminium) particles or magnetorheological (MR) fluids with ferromagnetic (iron) particles suspended in low-viscosity non-conducting silicone oil—aluminium may be extracted from anorthite minerals as described earlier while iron may be extracted from ilmenite minerals through hydrogen reduction at 1000 °C to iron metal and rutile (TiO_2_). The particles are ~0.1–100 μm in size. Application of high electric fields ~1–5 kV/mm increase the ER fluid viscosity by polarising the particles to form chain-like configurations [48]. The reverse process offers force-sensing capability. However, these approaches are based on carbon for both silicone side chains and carbon black, which must be sourced by heating regolith to release carbon volatiles.

Piezoelectric materials generate internal electric fields which alter their resonant frequency in response to mechanical stresses. The piezoelectric effect results from an electrical voltage change induced by an applied stress:(6)$$D_{i}=d_{ij}\sigma_{j}+\varepsilon_{ii}^{T}E_{i}=e_{ij}S_{j}+\varepsilon_{ij}^{T}E_{ii}$$
where D_i_ = electrical displacement, σ_ij_ = mechanical stress, ε_ii_ = electrical permittivity, E_i_ = electric field, S_j_ = mechanical strain, d_ij_ and e_ij_ = piezoelectric coefficients for a 3 × 6 piezoelectric matrix. Electrical displacement D defines the magnitude of the piezoelectric effect due to the applied mechanical stress σ and the electric field generated E according to the piezoelectric coefficients of the material d and e. Piezoelectric polymers such as PVDF can be used for inertial sensing and for tactile sensing [49]. A PVDF-trifluoroethylene film deposited on a MOSFET (metal oxide semiconductor field effect transistor) in conjunction with integrated temperature sensors constituted a tactile sensor array [50]. Embedding of organic FET arrays into silicone rubber measures pressure [51]. Organic thin-film transistors are p-type semiconductors based on conjugated polymers (such as pol(3-hexylthiophene-2,5diyl (P3HT) and 6,13 bis-(trisopropylsilylethynyl) (TIPS)-pentacene) that cannot be 3D printed into fine structures [52]. Such organics are not readily accessible due to the low incidence of carbon on the Moon, the complexity of their manufacture on the Moon and their poor tolerance to radiation.

PZT (lead zirconate titanate) is a common piezoelectric ceramic but there are other ceramic options—zinc oxide (ZnO), aluminium nitride (AlN), berlinite (AlPO_4_), topaz (Al_2_SiO_4_)(F,OH)_2_, barium titanate (BaTiO_3_), lead titanate (PbTiO_3_), etc. Ferroelectric materials such as PZT are also piezoelectric but not vice versa. Piezoelectric materials can be formed into semiconducting piezoceramic microwires that are embedded in silicone rubber to measure strain [53]. Piezoelectric ceramics offer much higher temperature tolerances than piezoelectric polymers, and embedding them in elastic polymers permits a degree of flexibility. However, these piezoelectric ceramics are scarce on the Moon.

The simplest and most widely available piezoelectric ceramic is quartz (SiO_2_), which is the second most abundant mineral on Earth after feldspar. On Earth, quartz is a major mineral of granite and it is the primary mineral constituent of sandstone. Quartz is scarce on the Moon however—maria basalts comprise only ~6% silica minerals (an example being cristobalite). Nevertheless, silica may be manufactured from lunar silicates (such as anorthite). HCl leaching of anorthite yields silicic acid from which silica is precipitated during the first stage of the two-stage production of alumina [15] (see Appendix A [22]). Quartz can be artificially synthesized from silica within a highly pressurized steel autoclave sealed with Bridgman seals—crystals are hydrothermally synthesized from a hot aqueous solution of melted silica below 573 °C. A temperature gradient is kept between the hot end that dissolves silica and the cool end that precipitates growing crystals of supersaturated quartz. Quartz may be used as the transduction material for force or pressure sensing or as a crystal actuator for radiofrequency oscillation. In the latter case, the simple Pierce oscillator circuit comprises only two resistors, two capacitors, one inverter and one quartz crystal. Piezoelectric tactile sensors, pressure sensors and a feedback circuit can output sensitive tactile measurements including differentiation between soft and hard objects by extracting phase shift in crystal resonance [54]. Quartz also provides the transduction component to the quartz microbalance (QMB) for the precise measurement of mass. All pyroelectric materials are piezoelectric including quartz which is sensitive to infrared radiation generated by temperature changes [55]. Piezoelectric force sensors are sensitive to dynamic forces but cannot measure static forces.

Active touch sensing involves using feedback-controlled behavior to actuate tactile sensors to maximize information gain [56]. Tactile precision is traded with the speed of movement. Tactility is an actuator-driven sensory modality because it is fundamentally exploratory. Sliding actuation generates ~micron-sized amplitude vibrations at ~200 Hz (a similar response to Pacinian corpuscles) that correlate well with textural roughness. The star-nosed mole *Condyllura cristata* possesses a nose that looks like a 22-fingered hand but acts like a tactile eye that is capable of recognizing prey using saccade-like movements of its foveal 11th appendage and then consuming the prey all within 120 ms [57]. Complex motor actuation patterns are required to generate a force distribution map using taxel arrays to form tactile “images”. The 3D location of the skin taxels must be calibrated through maximum likelihood mapping with respect to a central reference frame [58]. Tactile images may be subjected to edge and line detection algorithms to allow extraction of basic tactile properties using tactile moments such as contact area, centroid, eccentricity and principal axis. Tactile data is, however, noisy requiring the use of sophisticated filtering algorithms. Artificial skins commonly demonstrate poor tolerance to wear-and-tear with the requirement for integrated distributed electronics to which biological skin is robust and capable [59]. Actuation is a crucial component of sensation. There are distinctive actuation options in elastomeric skin. The Venus flytrap is a carnivorous plant that snaps its hinged lobes shut to trap insects. It can do this faster than hydraulic pressure due to its pre-stressed lobes switching between two stable mechanical states [60]. This is a binary contact switch which is simple but limited. In general, strain gauges and silicone elastomers/oils manufactured from lunar resources provide the basis for a plausible route to tactility.

## 3. Photomultiplier Tube (PMT)

Cameras are essential for all spacecraft and robotic operations offering the versatility of observability. On spacecraft, cameras may be used for self-visual monitoring of spacecraft state rather than relying on indirect measurements [61]. AERCams (autonomous EVA robotic cameras) are freeflying teleoperated cameras to support astronaut operations onboard the International Space Station [62]. Cameras will be ubiquitous for all lunar operations including the robotic construction of lunar infrastructure. Visible camera imaging is the primary sensing-at-a-distance measurement to support mobility [63]. A raw visual image comprises an array of light intensity values measured by each photosensitive pixel of the imaging array. First, we address individual pixels. There are several potential implementations of photosensitive pixels, photovoltaics being the most mature. However, precision doping to create pn junctions for photovoltaic pixels is too challenging to implement under lunar conditions [10,23]. There are other options offering high photosensitivity that would be highly desirable. Colloidal quantum dots constitute semiconducting fluorescent nanocrystals < 20 nm diameter synthesized through wet chemistry—a CdSe/TiO_2_ inorganic core is surrounded by a PMMA organic ligand [64,65]. Quantum confinement of quantum dots generates quantized energy levels that may be tailored but their efficiencies are currently low ~7% [66]. Pixels based on quantum dots are not feasible on the Moon for several reasons. First, sourcing and extracting Cd from lunar resources and manufacturing complex organic material would be too challenging, though TiO_2_ may be extracted from lunar ilmenite. Secondly, the high-precision microtechnology-based manufacturing required for quantum dots is even more challenging than that for photovoltaic cells. Given these limitations, we need to identify a suitable pixel form that can be constructed from available lunar resources and lunar-suitable manufacturing technologies such as 3D printing which are typically resolution-constrained.

Vision, like tactility, has enormous utility and ubiquity for agents, biological or robotic. It is to multicellular organisms that we look for bio-inspiration. There are a wide variety of different multicellular eye designs, most involving refractive lenses [67]. Mirror eyes of crustaceans such as lobsters and deep-sea fish such as the brownsnout spookfish are not refractive but reflective [68]. The lobster compound eye of ommatidia comprises a square corneal lens formed by a long crystalline pyramid with an axially decreasing refractive index acting as a set of mirrors for internal reflection. The brownsnout spookfish eye has a reflective layer comprised of high refractive index plates arranged with graded tilt angles to form a parabolic reflector. This is similar to the photomultipler tube (PMT) which may be configured into microchannel plate arrays similar to lobster eyes.

PMTs are vacuum tubes that operate via the photoelectric effect rather than through thermionic emission traditionally associated with vacuum tubes [69]. A PMT comprises a glass or ceramic enclosure housing a high vacuum. Within the tube is a photocathode and an anode sandwiching a series of dynodes. A transparent window to the cathode may be constructed from fused silica glass (which is manufactured from silica extracted from lunar silicates such as anorthite—see Appendix A). Fused silica glass is transparent to UV light up to 160 nm wavelength. The window focusses light onto a photo-emissive cathode. It emits electrons via the photoelectric effect that are accelerated by focussing electrodes through a series of dynodes (typically of around 10 stages). The dynodes are electron multipliers that amplify electron flux through secondary electron emission—there are typically ~4–6 secondary electrons emitted per incident electron for a few hundred volts. The dynode material is a secondary electron emitter that ejects electrons at energies in excess of the Fermi level and work function of >10 eV. Alkali earth metal oxides such as Al_2_O_3_ or MgO coatings on nickel, aluminium or steel dynodes are typical secondary electron emitters [70], though any alkali metal oxide (e.g., CaO or K_2_O) has secondary electron emission properties. All aforementioned oxides are derivable from lunar resources as delineated in our lunar industrial ecology [22]—Al_2_O_3_ from anorthite, MgO from olivine, CaO from anorthite and K_2_O from orthoclase (Appendix A). Stray magnetic fields can be mediated using permalloy (Ni-80/Fe-20) magnetic shielding. Both are extractable from nickel-iron (M-type) asteroid-derived resources. Each dynode is held ~100 V more positive than earlier dynodes to force the electrons to flow in one direction. Each stage adds ~100 keV of energy until >1–2 kV is reached at the anode. This generates an avalanche current of ×10^8^ amplified electrons. The quantum efficiency of the photoemissive material is defined as the ratio of output electrons to input photons:(7)$$\eta =(1-R)\frac{P_{\nu }}{k}\left(\frac{P_{s}}{1+1/kL}\right)$$
where R = reflection coefficient, P_ν_ = probability that absorbed light excites electrons to escape, k = photon absorption coefficient, P_s_ = probability that electrons are released from dynodes, and L = mean escape length. This is a property of the photoemissive material as the sole design parameter.

Photoelectron emission occurs when incident photon energy exceeds a threshold. This threshold is determined by the valence-conduction bandgap and the work function. This is a property of the photocathode material, usually, an alkali metal or III-IV semiconductor. Commonly adopted photoelectric materials include Cs_3_Sb and Na-K-Sb-Cs for visible and UV to NIR responses respectively. These cannot be manufactured from bulk lunar resources. Crystalline silicon has an indirect bandgap so it cannot be harnessed as a photosensitive transducer without dopants. The photocathodes can be constructed from another alkali metal with a low work function. A thin layer of K coated onto a W metal substrate exploits K’s work function of 2 eV—K may be sourced from lunar orthoclase. Aluminium in transmission mode has a slightly higher work function of 4.08 eV but requires very thin layers ~20 nm [71].

Semiconductor junction doping will be difficult to achieve on the Moon—the microwave applicator [72] is one option but its precision is not characterized. For optical sensitivity, we have chosen the simplest light-sensitive element, Se, as the photocathode. Se was the transducer in Alexander Graham Bell’s photophone (1880) to detect light modulated by sound-vibrated mirrors. Se powder is a p-type semiconductor with an energy gap of 1.99 eV in the visible waveband [73,74]. Photoelectric current is quantified by Fowler’s law: $i=k{\left(h\upsilon -\phi \right)}^{n}$ where k = constant, n = material exponent. On Earth, Se is sourced from the minerals clausthalite (PbSe), eucairite (CuAgSe) and crooksite (CuThSe) that occur in ores of metal sulfide. Hence, most Se on Earth is purified as a byproduct of electrolytic refining of chalcopyrite-rich (CuFeS) ores (where Se is substituted for S). Chalcopyrite does exist on the Moon but is scarce. Se itself is also rare on the Moon but it occurs in meteorites in an approximately constant Se/Te abundance ratio of ~10–20 by mass, the average Se content varying over 0.5–10 ppm [75]. In carbonaceous chondrites, Se is found as a substitution element to the 2500 times more abundant S. Troilite (FeS) is common in NiFe meteorites and is associated with graphite grains. Se can be released through HF/HNO_3_ leaching of sulfides in the presence of alumina (Al_2_O_3_). This is followed by purification with ascorbic acid [76] but organic acids will be difficult to synthesize on the Moon. Alternatively, we can adapt the treatment for CuSe to FeSe in troilite. Troilite is smelted with soda Na_2_CO_3_ in solution using a saltpeter (KNO_3_) catalyst at 650 °C (see Appendix A):
FeSe + Na_2_CO_3_ + 1.5O_2_ → FeO + Na_2_SeO_3_ + CO_2_

Sodium is scarce on the Moon so it must be imported from Earth (in the form of NaCl)—however, it is used as a reagent that is recycled and not consumed. Lunar orthoclase is a more practical source of K for saltpeter than KREEP minerals and nitrogenous lunar volatiles provides a source of N (Appendix A). Saltpeter may also be used for blasting with saltpeter mixed with sulfur and charcoal to form gunpowder. Troilite smelted with soda results in selenite Na_2_SeO_3_ which may be treated with H_2_SO_4_ to yield selenous acid (H_2_SeO_3_). Se may be precipitated from H_2_SeO_3_ at 700 °C:Na_2_SeO_3_ + 2H_2_SO_4_ → H_2_SeO_3_ + Na_2_O + 2SO_2_ +H_2_O → Na_2_O + 2H_2_SO_4_ + Se

Thus, the sulfuric acid reagent is recycled. This is a summary version of a multi-stage chemical reaction process:Na_2_SeO_3_ → Na_2_O + SeO_2_
SeO_2_ + H_2_O → H_2_SeO_3_
H_2_SeO_3_ + SO_2_ + H_2_O → 2H_2_SO_4_ + Se

Treatment of troilite (FeS) in the presence of H_2_S in an aqueous solution yields iron pyrite (FeS_2_) with the evolution of hydrogen. The reaction rate increases with temperature up to 125 °C [77]. A similar low-temperature transformation has been hypothesized to be part of a biotic redox couple powering early Earth’s iron-sulfur world [78]. Thin iron pyrite (FeS_2_) films ~μm thick are n-type semiconductors that are photoconductive with a high absorption coefficient α > 5 × 10^5^/cm for λ < 900 nm with a bandgap E_g_ = 0.95 eV for infrared sensitivity [79]. Thin film manufacturing by chemical vapour deposition on the Moon, however, remains an open question. Although we have examined the raw materials for constructing PMTs, we have not considered the manufacture of PMTs which are traditionally manufactured in parts and then assembled.

PMTs are optical detectors with high sensitivity and signal-to-noise ratio. PMTs may be arrayed into the pixels of a microchannel plate. Each parallel glass channel in this thin array are ~10 μm in diameter with ~15 μm spacing separated by electrically resistive walls. The glass walls of each channel act as continuous dynode electron multipliers onto which any photon striking causes a cascade of electrons. A strong electric field accelerates the electrons through each channel and amplifies the incident photons by many orders of magnitude. A PMT array may thus be deployed as a camera imaging array. Microchannel plates require microtechnology-scale manufacturing which is not feasible on the Moon. Given the bulkiness of lunar-manufactured PMTs, each pixel will have lower spatial resolution compared with microchannel plates or electronic cameras and the arrays will thus be modest in size. This limitation must be addressed, and we do so through bio-inspiration.

## 4. Bio-Inspired Vision

Traditional image processing is premised on large arrays of pixels of high resolution. The algorithms are computationally intensive and are unsuited to processing small PMT array data. We shall investigate biological vision to propose biomimetic approaches to robot vision to compensate for the low resolution and other challenges imposed by lunar-derived imagers.

Directed eye movements provide the basis for foveated vision. In fact, eye fixations are partly non-Bayesian in that they concentrate on high information with a large number of fixations within a narrow field of view (FOV) rather than pure random search with uniform distribution of fixations over a wider FOV [80]. Neural fields can rapidly learn mappings between retinal image space and the six-eye muscle motor space for driving visual saccades through random motor babbling [81]. A visual control policy of visual feature-to-action mappings may be learned through reinforcement (such as temporal difference learning) after the application of a Markovian visual classifier [82]. It is more efficient to compute different visual properties independently prior to integration during later processing mediated by synchronous firing. A similar separation occurs in the auditory cortex with independent “what” and “where” streams that are subsequently integrated during hearing spoken language [83,84]. We suggest that sensory data should be partitioned into independent processing streams—a vision chip with two complementary visual pathways implements this philosophy [85]. We propose separating the foveated vision (what) process for object identification and the optical flow (where) process for navigation to ease computational overheads.

## 5. Foveated Vision

The spatial resolution of lunar-constructed imagers comprising a small array of PMT pixels will be deficient, but there are biomimetic lessons to compensate for such limitations. Foveated vision implements actuator-driven exploration of the visual field. The human eye has ~7 × 10^6^ retinal cones packed at 5000 cones/μm within the ±0.5–1.0° fovea for high-resolution imaging [86]. The ~5° blindspot is the region where the retina projects into the optic nerve but it is invisible to perception. Away from the fovea, the other 120 × 10^6^ cones of the retina become more diffusely distributed, giving lower resolution for the rest of the visual field. Around 50% of the visual cortex is devoted to processing foveal data (the visual cortex itself comprises 50% of the cortex).

Gaze shifting and gaze holding allows rapid aiming of this narrow FOV as the basis of foveated vision [87]. Gaze shifting involves saccades while gaze holding constitutes visual fixations. Foveated vision may be characterized as a continuous sequence of visual fixations on specific but different visual field targets separated by saccades that propel the eyeball between these fixations. Successive saccades direct the eye to sample salient features as visual experiments that confirm prior hypotheses by minimizing prediction errors [88]. Movement complicates this simple description, but compensating mechanisms serve the purpose of maintaining visual fixation—smooth pursuit involves a moving fixation target (gaze following), the speed of which is limited by optokinetic reflex (OKR) but extended by vestibular-ocular reflex (VOR).

Saccades are fast eye movements that direct the fovea between different targets in the visual field. Foveated gaze control fixes the fovea on specific targets with visual fixations ~30 ms in duration. Saccades move point-to-point at 900°/s to bring the fovea to visual targets at a rate of 2–3 Hz amounting to ~100,000 searches per day. The saccades are controlled by a neural circuit through the frontal lobe, basal ganglia, superior colliculus and cerebellum. Between saccades, a neural integrator sustains an equilibrated eye position during visual fixation [89]. Every saccade changes the foveal direction of motion so tracking must be updated. This Bayesian approach computes the posterior likelihood as the prior likelihood updated by the latest fixation [90]. To reduce uncertainty, viewpoint selection for gazing follows the gradient $\frac{{\partial \sigma }^{2}}{\partial x}$ of the predicted variance σ [91]. The superior colliculus hosts a topographic map of saccade vector fields. This may be represented by a neural network with an upper layer connected to the lower layer by feedforward connections whose weights are determined by a recurrent backpropagation algorithm [92]. In the superior colliculus, reciprocal inhibition of gaze-shifting neurons or gaze-holding neurons determines which are active. Saccade is determined by the error between the current eye position and the desired eye position [93]. This influences neuronal discharge rates which are modulated through the time delay imposed by reaction times [94].

Foveated vision reduces optical hardware by orienting a narrow high-resolution fovea over the visual field. We may exploit such foveated vision in cameras with limited FOV mounted on a pan-tilt unit that slews the camera. This permits Gibsonian affordances which are potentials for action dictated by objects, events and locations in the environment relative to the agent [95,96]. Affordance is an ecological approach to cognition whereby perception of the environment is determined by actions on it which in turn determines perception, i.e., perception is an active process. In *Drosophila* fruitflies, visual features encoding object location in retinal coordinates are directly converted into action coordinates in body coordinates through synaptic weight gradients of topographically configured visual projection neurons (VPN) [97]. Slewing of cameras is executed by electric motors with rotary position feedback from rotary potentiometers at each motorized joint. The extended Kalman filter with iterative adaptation may be applied to the dynamics of such visual servoing [98].

There are two types of feedback for camera slewing—optical and vestibular. Optical flow comprises an optical flowfield indicating movement away from a focus of expansion in the environment. Foveated vision with its eye movements with respect to the body add supplemental retinal image motion to be superimposed on the flowfield in the direction of gaze. Eye movements shift the focus of expansion by (d/x)θ where d = distance of the place from the observer, θ = rotation of the observer, and x = translation of the observer. OKR stabilizes a moving image on the retina by measuring retinal slip as the image moves. There is a latency of 80–100 ms to retinal slip feedback. Visual estimation of movement through optical flow is sufficient for slow eye movements < 1.5°/s (OKR) but, for higher speeds, vestibular information from the VOR is required.

VOR uses feedback on head movement from the vestibular organs in the semicircular canals to stabilize gaze. There is a much shorter latency of 15–30 ms to vestibular feedback. As VOR responds only to head acceleration, constant velocity head movement invokes OKR. Both VOR and OKR are mediated by the cerebellar flocculus. Stabilization of gaze through VOR may be implemented through feedback error learning with a neural network forward model [99]. The feedforward controller utilizes more rapid vestibular feedback gaze stabilization than vestibular feedback alone to eliminate blurring. In a robotic implementation for a mast-mounted camera, the feedforward model may be adapted to accommodate predicted vestibular states using proprioceptive (motor) data from the deploying manipulator [100]. The unscented Kalman filter can implement VOR-inspired visual servoing [101]. VOR is a reflex that stabilizes images on the retina by implementing compensating eye movements using measurements by the semicircular canals during head movement. Signals from the semicircular canals are transmitted rapidly via a three-neuron arc with a time lag of only 10 ms to the eye muscles. VOR is a feedback system with gain adaptation to facilitate the integration of several information sources [102,103].

Translating this to a pan-tilt camera assembly, inertial measurement of the camera itself is not commonly adopted. A bio-inspired approach exploits feedback from the joints of the camera’s mast/pan-tilt unit. In mammals, feedback from muscles is generated by spindles (position-derivative) and Golgi tendon sensors (force) to provide muscular force feedback to direct the eyes. Predictive feedforward control can augment feedback to compensate for reduced observability and thereby generate robust visual tracking [100]. We have implemented this forward model on a robotic manipulator-mounted camera to demonstrate smooth pursuit of a moving target. Conventionally, the camera platform itself does not incorporate inertial sensors such as gyroscopes/accelerometers. Forward modelling of camera attitude using a neural network, substitutes for this lack of sensory data. This feedforward model enhances feedback from rotary position sensors of the pan-tilt joints (such as rotary potentiometers) and actuated camera pointing to reduce error excursions from commanded camera pointing during smooth pursuit (Figure 2).

VOR is suppressed by smooth pursuit to eliminate conflicting interference in eye movements. The Kalman filter that merges noisy sensory inputs with a predictive model effectively reproduces biological smooth pursuit behavior of eye velocity under pure visual to pure predictive conditions [104]. The Kalman filter configuration comprises two Kalman filters, one for processing visual data to estimate retinal slip (in area MT), and the other implementing an internal predictive dynamic model of object motion for 150 ms into the future (in frontal eye field), each optimally weighted according to their reliability.

Camera motor control is core to foveated vision. We have demonstrated that DC electric motors may be potentially 3D printed from lunar resources [105]. Our 3D-printed motor prototype was printed in Proto-Pasta^TM^ (Protopasta, Vancouver, WA, USA) comprising 50% iron particles embedded in a 50% PLA (polylactic acid) matrix by mass. This constituted the soft magnetic material of the motor (Figure 3).

The closed magnetic circuit stator and the rotor were constructed from Proto-Pasta^TM^ as the soft magnetic components. This is a dual excitation motor where the copper wire was wound around both to create a fixed electromagnetic stator and an alternating electromagnetic rotor respectively. A 3D-printed lunar variant could replace Proto-Pasta^TM^ with iron particle-impregnated fused silica glass or nanophase-iron-impregnated lunar glass for the stator/rotor. Copper windings may be replaced with nickel, kovar or aluminium windings for a lunar version. For a permanent magnet motor, permanent magnets of AlNiCo may be potentially manufactured from lunar resources [106]. The main structure of a lunar motor may be manufactured from lunar glass. Such motors would be the primary mechanism for implementing foveated vision. Feedback control of the joints may be implemented using rotary potentiometers constructed from metal (aluminium or nickel), ceramic (metal oxide or glass) or cermet resistors. Potentiometers offer a higher resolution of control ~1° than PMT pixel resolution allowing a camera to orient its FOV at higher-than-pixel resolution. A stepper motor variant of this 3D-printed motor (with more stator poles) can improve this resolution substantially through micro-stepping.

## 6. Optical Flow Vision

While foveated vision performs object identification, optical flow performs visual navigation. Optical flow is integral to mammalian vision—even during eye fixations, there are involuntary microsaccades of ~arcminute amplitudes and ~0.5 s periods. Microsaccades are necessary for vision as immobilization of the eyeball cause objects to disappear. The retina detects relative intensity fluctuations rather than absolute intensity which yields visual artifacts such as shadowing by blood vessels, etc. Gibsonian ecological constraints on the environment limit potential interpretations of sensory stimuli [96]. Such stimulus information is encoded as invariants in the environment—optical flow is such an affordance and requires minimal inferential processing [107].

Flying insects exploit optical flow because their eyes have fixed orientation and fixed focus with respect to their bodies [108]. The small baseline between their eyes does not typically allow binocular stereopsis for depth estimation (though there are exceptions). Optical flow generates information on both self-motion and that of objects in the environment. Insects possess brains of only ~10^6^ neurons to perform low-level visuomotor processing. Spatial filtering is implemented by low pass Gaussian filters on photoreceptor signals while temporal high-pass filtering is implemented through predictive coding. *Drosophila melanogaster*‘s eye comprises a 2D array of 700 ommatidiae with overlapping Gaussian receptive FOV and variable spatial resolution over the eye [109]. Ommatidia form geometric groups with a central pixel surrounded by six neighboring pixels. Motion is detected by comparing visual signals in the central pixels with delayed signals in neighboring pixels. The overlapping Gaussian response of the ommatidia gives it superior performance. Behind the ommatidiae are three neural layers—neuropils lamina, medulla and lobula complex. These layers perform contrast enhancement, signal amplification and motion detection. Comparing pixel intensities in the reference patch of the visual field with delayed pixel intensities in neighboring patches extracts motion.

An elementary motion detector (EMD) such as a Reichardt detector generates its strongest response when a visual pattern moves in a specific direction [110]. EMD arrays can model insect compound eyes of 3000 pixels [111]. Object velocities are measured between neighboring facets to generate a polar map of obstacles with respect to an eye-centred polar reference frame. An insect compound eye has been modelled with an array of 100 analogue Reichardt detectors on a mobile robot [112]. A PI controller can reproduce the summation process of neighboring neurons [113]. A compound eye bio-inspired from the bee has been photolithographically constructed from a hemispherical array of ommatidia, each comprised of a hexagonal refractive PDMS microlens to focus light onto a PDMS cone through a refractively cladded waveguide of photosensitive resin to collect light from a narrow FOV onto a photodetector [114,115]. The ommatidia point radially to give a wide FOV. A bio-inspired camera based on a housefly’s compound eye exhibited superior performance to a commercial CCD camera in detecting moving objects under low contrast [116]. The biomimetic sensor was based on a neural superposition compound eye rather than an apposition compound eye such as those modelled by Reichardt detectors. The biomimetic sensor exhibited 70% overlap between Gaussian receptive fields of ommatidia separated by 4.5°.

Insects use image motion to estimate range such that images of nearer objects move faster than those of objects further away [108]. Hence, distance may be estimated by time-integrating image as optical flow is inversely proportional to distance due to translational motion. Rotational motion generates equal angular motion at all distances. Hence, translation and rotation motion can be distinguished when at least six points can be tracked. Optical flow speed v determines object distance d:(8)$$d=\frac{v}{w}{sin}^{2}\theta$$
where d = closest approach distance to an obstacle, v = velocity of the agent with respect to the object, w = angular velocity of the agent, θ = angle to the object. Hence, faster optical flow of an object across the visual field indicates a nearer object. Flying insects maintain equal left/right distance between passing obstacles by balancing retinal image angular speeds in the left and right eyes.

Avoidance of collisions is a fundamental part of sensorimotor control—the τ-hypothesis requires the computation of time-to-collision from visual stimuli [117]. During approach movements, the retinal image expands (looming) to give a measure for time-to-contact τ. A spherical object of diameter D approaching the eye along the line of sight at velocity v subtends an angle θ at a stance z from the eye such that $\theta =\frac{D}{z}$ and $\tau =\frac{z}{v}=\frac{\theta }{\dot{\theta }}$, i.e., the ratio of object image size to the rate of change of size of image expansion gives time-to-collision. It has been suggested that the tau-hypothesis is incorrect as visual stimulus does not contain sufficient information to operate effectively [118] and that binocular information is required from both eyes—a point moving with speed v towards the binocular eye midpoint yields:(9)$$\tau =\frac{x}{v}=\frac{sin\delta }{\dot{\delta }}$$
where δ = horizontal binocular disparity, x = distance to object. Some insects invoke self-motion to generate optical flow information [119,120]. Optical flow is a facility for robust visual navigation that may be implemented by simple electronic circuitry [121]. Optical flow sensors may be combined with foveated vision to adjust gaze orientation to maintain a parallel orientation with respect to local surface curvature [122]. This allows optical flow to be computed directly in the local environment reference frame. In essence, this is a form of peering that foveates vision to a specific target and simultaneously generates self-motion-induced image flow perpendicular to the optic axis [123]. Rather than treating foveated vision and optical flow as independent processing streams, their integration introduces the necessity for hierarchical information processing. A single four-layer neural network can integrate top-down object location (where) with bottom-up object identification (what) through a saliency map to control visual attention [124,125,126,127].

## 7. Conclusions

We have been concerned with the identification of raw material resources on the Moon and how to extract the desired refined materials for specific applications. We have shown that sensors can be manufactured from lunar resources with certain provisos. Such provisos include that we have not considered the full manufacture of these sensors from feedstock, which requires more detailed treatment [128]. Suitable sensors are those for measuring the most fundamental parameters for robotic machines. We suggest that potentiometers to measure displacement and resistance thermometers to measure temperature could be manufactured from lunar material. Quartz may be synthesized from silica extracted from lunar silicates—this provides a transduction mechanism for the measurement of pressure, mass and time. Similarly, PMTs may be manufactured from lunar resources to offer measurement of light intensity (vision). However, biomimetic lessons can compensate for PMT array limitations including foveated vision and optical flow navigation. Sensor technology is a crucial aspect of robotics in implementing the sensor-controller-actuator cycle. If full robotic capacities are to be manufactured from lunar resources, we must show that sensors (including vision), controllers and actuators can be manufactured as critical components. If this can be demonstrated, then robotic machines of production necessary for building lunar infrastructure can themselves be constructed from lunar materials. This paper represents a start in showing a potential path for manufacturing robotic sensors from lunar resources.

The methods we address and propose are not fait accompli—as in all things, the devil will be in the details. To be sure, showing the chemical processes does not address the engineering implementations. For example, the step from laboratory demonstration to practical realization in a lunar payload is a significant one. There are multiple considerations yet to be addressed including (but not exclusively): (i) feedback control of chemical processing which is inherently nonlinear and time-varying; (ii) material processing, handling and throughput plumbing between processing stations must accommodate transport of different forms of product and feedstock; (iii) automation of transport and throughputs of samples robotically including fault handling; (iv) manufacturing planning, allocation and monitoring for part fabrication; (v) at all stages, analytical instrument integration for monitoring of processing conditions and product integrity. In a broader context, we have explored how bio-inspiration from terrestrial organisms can yield novel technological solutions to the challenges of building an infrastructure on the Moon *de novo*. In particular, the *de novo* condition requires the construction of fundamental components (here, tactile and vision sensors) from local resources as the foundation on which to build organisation and hierarchies that enable more complex capabilities. A similar problem may have been encountered during the origin of life on Earth [19].

## Figures and Tables

**Figure 1 biomimetics-09-00496-f001:**
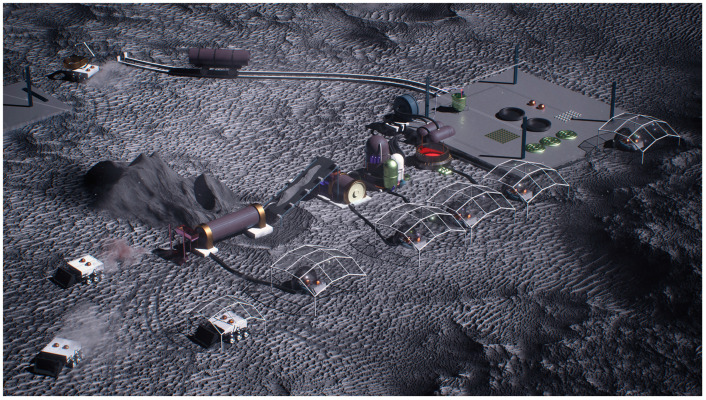
Artistic impression of a lunar industrial architecture for building lunar infrastructure.

**Figure 2 biomimetics-09-00496-f002:**
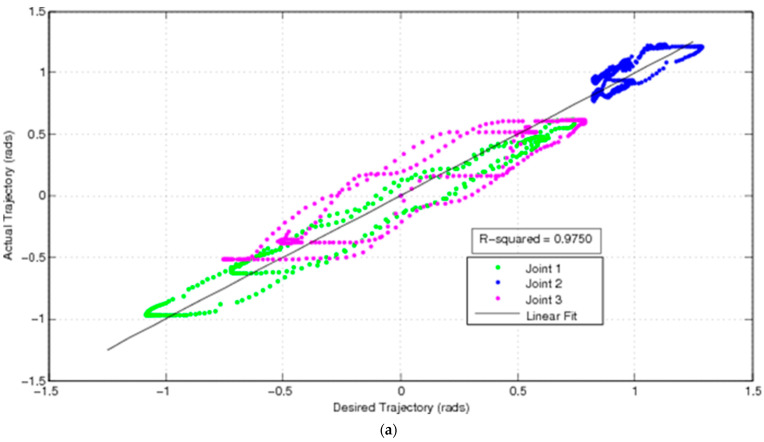
Error excursion of camera from its desired pointing trajectory (**a**) using feedback control alone and (**b**) using feedback supplemented by feedforward control (from [97]).

**Figure 3 biomimetics-09-00496-f003:**
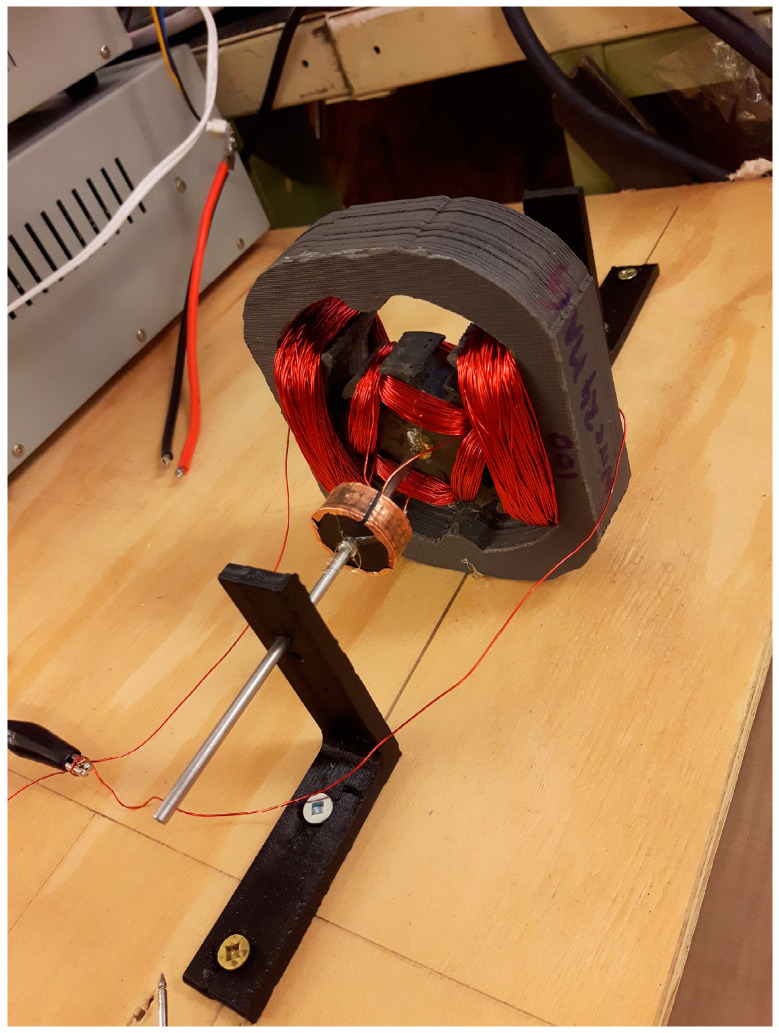
FDM-printed rotor and stator using Proto-Pasta^TM^: the rotor has a diameter of 50 mm by length of 15 mm embedded with the stator of width of 95 mm by height of 105 mm by length of 25 mm.

## Data Availability

No new data was generated for this article.

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
