# Peer review of "Bio-Inspired Strategies Are Adaptable to Sensors Manufactured on the Moon"

_biomimetics, 2024, doi:10.3390/biomimetics9080496_

Round 1
Reviewer 1 Report
Comments and Suggestions for Authors
Congratulations to the author on a very comprehensive and thorough paper. I found it to be scientifically sound and worthy of publication in its present form. I have been studying the Moon for many years yet I learned a lot from this paper. The organization, technical material, and references are very coherent.
With the planned near-term human exploration and settlement of our Moon, the methodology described here will become increasingly relevant. The manufacturing of devices needs to return to fundamental concepts preferably with bio-inspired strategies adaptable as appropriate.
As far as additional comments, I found that the paper nicely addressed how to make robotic sensors needed for future advanced lunar exploration can benefit from biological systems and biomimetic strategies. It went into a detailed description with examples of potential biological phenomena from which designers can develop ideas for spacecraft instrumentation with emphasis on adaptability and hierarchy.
This by itself is not original thinking, as reflected in the references, and is not necessarily a breakthrough leading to a specific invention. But it is an essential part of research to advance this field methodologically, which is why it is worthy of publication as an excellent contribution to the general literature in this field.
The author started with the basic idea of 3D printing of components in the lunar environment and developed a method for shape development of the final product in a manner similar to biomimetic polymer shaping, taking into account resource constraints when needed. Similarly, in another section, he describes the similarity between active touch sensing and feedback loops in actuation, and so on.
So, the paper, in its own words “This paper represents a start in showing a potential path for manufacturing robotic sensors from lunar resources.”
(no additional files to upload).
Author Response
Congratulations to the author on a very comprehensive and thorough paper. I found it to be scientifically sound and worthy of publication in its present form. I have been studying the Moon for many years yet I learned a lot from this paper. The organization, technical material, and references are very coherent.
With the planned near-term human exploration and settlement of our Moon, the methodology described here will become increasingly relevant. The manufacturing of devices needs to return to fundamental concepts preferably with bio-inspired strategies adaptable as appropriate.
RESPONSE: Thank you for your very positive comments!
As far as additional comments, I found that the paper nicely addressed how to make robotic sensors needed for future advanced lunar exploration can benefit from biological systems and biomimetic strategies. It went into a detailed description with examples of potential biological phenomena from which designers can develop ideas for spacecraft instrumentation with emphasis on adaptability and hierarchy.
This by itself is not original thinking, as reflected in the references, and is not necessarily a breakthrough leading to a specific invention. But it is an essential part of research to advance this field methodologically, which is why it is worthy of publication as an excellent contribution to the general literature in this field.
RESPONSE: Thank you!
The author started with the basic idea of 3D printing of components in the lunar environment and developed a method for shape development of the final product in a manner similar to biomimetic polymer shaping, taking into account resource constraints when needed. Similarly, in another section, he describes the similarity between active touch sensing and feedback loops in actuation, and so on.
So, the paper, in its own words “This paper represents a start in showing a potential path for manufacturing robotic sensors from lunar resources.”
RESPONSE: Thank you, again! It is clear the reviewer has understood the raison d’etre for this paper.
Reviewer 2 Report
Comments and Suggestions for Authors
Thank you for the submission of your manuscript. I have carefully reviewed your work, and after thorough consideration, do not believe I can recommend it for publication at this time. My decision is based on several critical factors outlined below:
-
Excessive Length and Discursive Style: The manuscript is extremely long-winded and discursive, making it challenging to follow the central arguments and ideas. The paragraphs are notably lengthy, which contributes to the difficulty in maintaining reader engagement and clarity of thought.
-
Lack of Focus on Lunar Applications: Although the paper purports to address the development of strategies for bio-inspired sensors suitable to lunar applications, the majority of the content is dedicated to describing examples of biological systems and their electromechanical equivalents. There are several paragraphs in section 2 that simply end by effectively saying "...but this isn't relevant to the Moon". Why, then, was it discussed at length? This detracts from the main theme of the manuscript and leaves a significant gap in demonstrating how these systems could be practically developed for lunar applications.
-
Inadequate Conclusion Alignment: The first sentence of the conclusion states, "We have shown that sensors are manufacturable from lunar resources with certain provisos." However, this claim is not sufficiently supported within the body of the manuscript. Only a small portion of the paper appears to be dedicated to this (see point 4), weakening the overall impact and coherence of the study. It is inaccurate and misleading to say something has been "shown" when the author has simply said that something is technically possible, but glosses over many of the associated challenges with doing so (see point 5).
-
Overemphasis on Biological Knowledge: It appears that the manuscript leans more towards showcasing the author's knowledge of biological systems rather than focusing on the primary objective of developing strategies for bio-inspired sensors for lunar applications. This misalignment with the stated objectives further complicates the manuscript's narrative and its suitability for the journal.
-
Practical Considerations Overlooked: The discussion on the fabrication of, for example, synthetic quartz and other materials suggests a lack of consideration for the practicalities involved. While the fabrication of such materials may be possible, the requirement for advanced manufacturing equipment (technologies of a complexity far greater than that of the sensors being discussed) and the feasibility of such processes in a lunar environment are not adequately addressed.
Author Response
Thank you for the submission of your manuscript. I have carefully reviewed your work, and after thorough consideration, do not believe I can recommend it for publication at this time. My decision is based on several critical factors outlined below:
- Excessive Length and Discursive Style: The manuscript is extremely long-winded and discursive, making it challenging to follow the central arguments and ideas.
RESPONSE: 29 pages is not an excessive length for a review paper – I have seen far longer review papers in engineering. No specific examples of “long-winded and discursive” style were offered indicating how it was difficult to follow. This comment may be regarding the introductory material that elicited confusion because it served three functions: (i) to introduce the lunar in-situ resource utilisation context which is likely unfamiliar to most readers of Biomimetics; (ii) an exposition on the subject-matter of the special issue – a systems approach – by elucidating on how biological systems required the evolution of biological components; (iii) how the design of said components are bio-inspired. These issues were eluded to throughout the paper as a constant thread. However, I have removed section 7 to shorten the paper.
The paragraphs are notably lengthy, which contributes to the difficulty in maintaining reader engagement and clarity of thought.
RESPONSE: I have split the longer paragraphs into smaller one at lines 173, 208, 488, 584, 705, 735, 760, 839 to make it easier for the reader.
- Lack of Focus on Lunar Applications: Although the paper purports to address the development of strategies for bio-inspired sensors suitable to lunar applications, the majority of the content is dedicated to describing examples of biological systems and their electromechanical equivalents. There are several paragraphs in section 2 that simply end by effectively saying "...but this isn't relevant to the Moon". Why, then, was it discussed at length? This detracts from the main theme of the manuscript and leaves a significant gap in demonstrating how these systems could be practically developed for lunar applications.
RESPONSE: I added through the text further expositions on ISRU which are highlighted in red including more references on lunar ISRU for further information (to discuss them at length would add further length to the paper and the material is already published). I can find only one paragraph (actually, 5 lines) where it is explicitly stated “not relevant to the Moon” after a discussion - lines 302-307 describing artificial cilia because it requires lithographic manufacturing. Excluding options is part of any engineering design process so it is legitimate to discuss it. I then proceed to discuss larger whiskers pro sequitur. There are other cases where I’ve stated irrelevance to the Moon but this is not discussed at length (e.g. solid state electronics). These are also engineering design decisions.
- Inadequate Conclusion Alignment: The first sentence of the conclusion states, "We have shown that sensors are manufacturable from lunar resources with certain provisos." However, this claim is not sufficiently supported within the body of the manuscript. Only a small portion of the paper appears to be dedicated to this (see point 4), weakening the overall impact and coherence of the study. It is inaccurate and misleading to say something has been "shown" when the author has simply said that something is technically possible, but glosses over many of the associated challenges with doing so (see point 5).
RESPONSE: “with provisos” qualifies the statement. I have now preceded the statement to make this clear with Line 949: “We have been concerned with the identification of material raw resources on the Moon and how to extract the desired refined materials for specific applications.” I have added the statement to further clarify this with Line 951: “Such provisos include that we have not considered manufacturing these sensors from feedstock which requires more detailed treatment” with a reference that does consider such. Line 975 clearly states “This paper represents a start in showing a potential path for manufacturing robotic sensors from lunar resources.”
- Overemphasis on Biological Knowledge: It appears that the manuscript leans more towards showcasing the author's knowledge of biological systems rather than focusing on the primary objective of developing strategies for bio-inspired sensors for lunar applications. This misalignment with the stated objectives further complicates the manuscript's narrative and its suitability for the journal.
RESPONSE: The exposition on the biological aspects and engineering realisation is the core of the paper as appropriate for a Biomimetics journal. As a paper for a Biomimetics journal, biological discussion is both a requirement and a necessity. To remove all the biological discussion would render the paper out-of-scope for the journal which is an odd suggestion to improve “its suitability for the journal”. Furthermore, the special issue on “Systems Approach” necessitated discussion of relevant biological issues including making comparisons between biology and engineering. This is a review paper where such discussions are entirely suitable. The bio-inspired approach presented is quite clear especially in the case of vision which constitutes the bulk of the discussion. Line 191: “Lunar resource availability imposes performance limits…..but bio-inspired approaches can compensate…”. The adoption of PMTs introduces the requirement for subpixel resolution – this is enabled by foveated vision. The adoption of parallel processing streams permits specialised processing. The logic is clear. There is no misalignment.
- Practical Considerations Overlooked: The discussion on the fabrication of, for example, synthetic quartz and other materials suggests a lack of consideration for the practicalities involved. While the fabrication of such materials may be possible, the requirement for advanced manufacturing equipment (technologies of a complexity far greater than that of the sensors being discussed) and the feasibility of such processes in a lunar environment are not adequately addressed.
RESPONSE: The reviewer did not suggest what “advanced manufacturing equipment…[with]…complexity far greater than that of the sensors” that they were thinking of. Lines 372-376 describe the requirements for the production of quartz – it requires a sealed pressure vessel but modest temperatures with pressure and temperatures sensors. This is not more complex than a PMT.
In general, I have clearly defined the scope of the article in several places:
Line 217 clearly states “As we focus on material availability….investigation includes feasible chemical processing pathways….”
Line 227 clearly states “roadmap to manufacturing….emphasis on lunar materials and aspects of their chemical processing rather than manufacturing techniques….”
Line 231 states “We are examining resource restrictions aspect of manufacturing sensors….”
Line 582 states explicitly that manufacturing of PMTs is not within scope of the article.
Round 2
Reviewer 2 Report
Comments and Suggestions for Authors
Thank you for the modifications you have made. I believe that there are still improvements that could be made to improve the manuscript prior to publication.

Author Response
RESPONSE 2 (General Clarification): All new changes are marked up in red. In addition, many references are in red – this is for my own benefit to enable me to re-number the referencing during final formatting.
Thank you for the submission of your manuscript. I have carefully reviewed your work, and after thorough consideration, do not believe I can recommend it for publication at this time. My decision is based on several critical factors outlined below:
- Excessive Length and Discursive Style: The manuscript is extremely long-winded and discursive, making it challenging to follow the central arguments and
RESPONSE: 29 pages is not an excessive length for a review paper – I have seen far longer review papers in engineering. No specific examples of “long-winded and discursive” style were offered indicating how it was difficult to follow. This comment may be regarding the introductory material that elicited confusion because it served three functions: (i) to introduce the lunar in-situ resource utilisation context which is likely unfamiliar to most readers of Biomimetics; (ii) an exposition on the subject-matter of the special issue – a systems approach – by elucidating on how biological systems required the evolution of biological components; (iii) how the design of said components are bio- inspired. These issues were eluded to throughout the paper as a constant thread. However, I have removed section 7 to shorten the paper.
Reviewer Reply: Certainly, the absolute length of the paper is not excessive for a review, however I found some of the content within this work to be excessive and not in support of the stated aims. This gives the impression of a piece that is too long for its intended purpose. Regardless, thank you for removing section 7 to shorten the paper.
RESPONSE 2: I have substantially restructured the Introduction so that it is tighter and more focused -see next Response for details.
I still find the introduction hard to follow, and somewhat repetitive. One example: line 118-119 reads almost identically to 214-215. I would suggest the author introduce sub-headings that align with the three functions described above in order to encapsulate discretely the discussion and improve the flow.
RESPONSE 2: I would like to thank the reviewer for these more detailed comments – on examination, you are correct, there was not only repetition but poor structure in the Introduction. I have entirely re-structured the Introduction by excising two paragraphs and dispersing them through the next section where relevant and merged the remaining paragraphs. The narrative should flow more smoothly now.
There is a point in the introduction (line 208) where it seems like the section has been concluded, however it continues on. The paragraph ending at 208 describes the structure of the paper, a natural finishing point for introductions, however there are two subsequent paragraphs that read as afterthoughts, and don’t conclude the section in a logical way. This needs to be restructured.
RESPONSE 2: Thank you for this clarification. The first of the two mentioned paragraphs has been merged with earlier text. All that remains now is a statement of scope (relating to your comments in round 1). The second paragraph (now merged with the above statement of scope) remains again as a clarification that many aspects of lunar operations reside outside the scope of the article.
Line specific comments:
65: ‘ball mill for comminution’ – Why has the author included reference to milling for lunar regolith? The median particle size is 72 µm, and 90% of the material is <1 mm. Comminution circuits will likely unnecessary for lunar ISRU for the foreseeable future.
RESPONSE 2: I am indeed aware of regolith particle size distribution. We have recently examined trades between traverse/scooping energy consumption demands imposed by sieving and discarding versus comminution energy in grinding the larger particle fraction (yet to be published). This suggests that processing efficiency favours milling larger regolith particles rather than discarding them for enhanced bulk material extraction efficiency. Nevertheless, I have removed the reference to ball-milling.
179-180: “…releases water vapour at 100°C…” – this would be true under atmospheric pressure, however in the hard vacuum on the lunar surface, this will happen at much lower temperatures. If the vessel is pressurised, this should be mentioned. Otherwise, the numbers should be corrected for the vacuum on the Moon.
RESPONSE 2: You are correct but a complicating factor occurs subsurface water ice is covered by lunar dust which imposes mechanical pressure under lunar gravity. Under hard vacuum, sublimation occurs above -53oC – I have corrected this (line 299).
241: “…froth flotation…” – flotation is an excellent beneficiation technology for terrestrial mining, but it is entirely impractical and unsuitable for lunar ISRU. This should be removed.
RESPONSE 2: You are correct about froth flotation – reference to it has been removed (line 277).
261-262: “…asteroidal material that may well be buried…” – There is nickel in the lunar regolith, however a quick scan of the literature shows that native nickel exists in the regolith as well as from impactors. I would suggest the author broaden the literature survey to include other works.
RESPONSE 2: Ni content in bulk regolith is very low ~50 ppm requiring processing of large amounts of regolith to extract even small volumes of Ni. Traversing and scooping regolith plus processing large bulk to extract the small amounts of Ni consumes much energy, so this places a premium on higher concentration sources of Ni. For this reason, sourcing subsurface NiFe ores with high Ni concentrations is highly desirable. Hence, our emphasis on this approach.
The paragraphs are notably lengthy, which contributes to the difficulty in maintaining reader engagement and clarity of thought.
RESPONSE: I have split the longer paragraphs into smaller one at lines 173, 208, 488, 584, 705, 735, 760, 839 to make it easier for the reader.
Reviewer Reply: Thank you, this improves the readability.
- Lack of Focus on Lunar Applications: Although the paper purports to address the development of strategies for bio-inspired sensors suitable to lunar applications, the majority of the content is dedicated to describing examples of biological systems and their electromechanical There are several paragraphs in section 2 that simply end by
effectively saying "...but this isn't relevant to the Moon". Why, then, was it discussed at length? This detracts from the main theme of the manuscript and leaves a significant gap in demonstrating how these systems could be practically developed for lunar applications.
RESPONSE: I added through the text further expositions on ISRU which are highlighted in red including more references on lunar ISRU for further information (to discuss them at length would add further length to the paper and the material is already published). I can find only one paragraph (actually, 5 lines) where it is explicitly stated “not relevant to the Moon” after a discussion - lines 302- 307 describing artificial cilia because it requires lithographic manufacturing. Excluding options is part of any engineering design process so it is legitimate to discuss it. I then proceed to discuss larger whiskers pro sequitur. There are other cases where I’ve stated irrelevance to the Moon but this is not discussed at length (e.g. solid state electronics). These are also engineering design decisions.
Reviewer Reply: I agree that there is always a point in the design cycle where certain technology exclusions will be made from the overall design space. However, this is not an engineering design paper. In the interest of keeping to the main point, I would suggest saying something like “Other systems, such as artificial cilia, that would work in principle, but are unsuitable for manufacturing on the Moon due to process constraints”, or something to that effect. It isn’t relevant to go into depth on a particular topic if it isn’t suitable; it serves no purpose other than to increase the length.
RESPONSE 2: During this type of early development, the design cycle is not fully structured. Aspects of engineering design have already been undertaken (such as our lunar industrial ecology) but not completed – this provides context for the paper. The elimination of cilia makes an important point that I have emphasised several times – solid state technology cannot be implemented on the Moon for the foreseeable future and this has significant ramifications including elimination of cilia as a consideration. But whiskers are enlarged cilia….I think it important to mention this and it takes only 5 lines – even a cursory statement on cilia would likely take 2 lines thereby saving only 3 lines….
- Inadequate Conclusion Alignment: The first sentence of the conclusion states, "We have shown that sensors are manufacturable from lunar resources with certain provisos." However, this claim is not sufficiently supported within the body of the manuscript. Only a small portion of the paper appears to be dedicated to this (see point 4), weakening the overall impact and coherence of the study. It is inaccurate and misleading to say something has been "shown" when the author has simply said that something is technically possible, but glosses over many of the associated challenges with doing so (see point 5).
RESPONSE: “with provisos” qualifies the statement. I have now preceded the statement to make this clear with Line 949: “We have been concerned with the identification of material raw resources on the Moon and how to extract the desired refined materials for specific applications.” I have added the statement to further clarify this with Line 951: “Such provisos include that we have not considered manufacturing these sensors from feedstock which requires more detailed treatment” with a reference that does consider such. Line 975 clearly states “This paper represents a start in showing a potential path for manufacturing robotic sensors from lunar resources.”
Reviewer Reply: The additional clarification is helpful.
- Overemphasis on Biological Knowledge: It appears that the manuscript leans more towards showcasing the author's knowledge of biological systems rather than focusing on the primary objective of developing strategies for bio-inspired sensors for lunar applications. This misalignment with the stated objectives further complicates the manuscript's narrative and its suitability for the
RESPONSE: The exposition on the biological aspects and engineering realisation is the core of the paper as appropriate for a Biomimetics journal. As a paper for a Biomimetics journal, biological discussion is both a requirement and a necessity. To remove all the biological discussion would render the paper out-of-scope for the journal which is an odd suggestion to improve “its suitability for the journal”. Furthermore, the special issue on “Systems Approach” necessitated discussion of relevant biological issues including making comparisons between biology and engineering. This is a review paper where such discussions are entirely suitable. The bio-inspired approach presented is quite clear especially in the case of vision which constitutes the bulk of the discussion. Line 191: “Lunar resource availability imposes performance limits…..but bio-inspired approaches can compensate…”. The adoption of PMTs introduces the requirement for subpixel resolution – this is enabled by foveated vision. The adoption of parallel processing streams permits specialised processing. The logic is clear. There is no misalignment.
Reviewer Reply: To clarify, I was not proposing the removal of all biological aspects from the paper. That would, as the author states, be inappropriate. Moreso, my intention was suggest that there are aspects of the paper that read as if they’ve been included for the sake of bulking up the biological credentials of the paper without adding a great deal of value. One example, lines 427-429 (discussion of the star nosed mole), is obviously related to the preceding and subsequent sentences, but the connection is not clear, and comes across as having been injected into a paragraph during a round of revisions. Another example, lines 469-500. These two paragraphs provides a great deal of context to the origin of vision systems in biology. While it is interesting, I fail to see how knowing the context of vision in cells helps to build the argument for bio-inspired sensor design. It provides background and context, sure, but to me, doesn’t add to the discussion. Rather, it reads as an unnecessary digression.
RESPONSE 2: It is these biological aspects – they are not digressions - that render the paper relevant to the special issue. This is a Biomimetics journal in which the biological aspects are of the most interest to the journal readership. The special issue is concerned with biological systems and their engineering manifestation. As an engineer, I find the biological aspect instructive as well as inspirational and not at all irrelevant. The star-nosed mole segues from tactility to vision, illustrating fluidity between sensing modalities. The material on biological vision illustrates that vision is fundamental and evolved in single-celled organisms. This brings the discussion back to biomimetics and the basis of biological hierarchical systems (special issue relevance) – visual components lie at the base of biological hierarchy. Visual (and tactile) components are the topic of the paper. A discussion of types of biological eyes leads us to the brownsnout spookfish eye which is a biological analogue of the PMT. Without these discussions, the paper would have diminished relevance to the special issue topic.
- Practical Considerations Overlooked: The discussion on the fabrication of, for example, synthetic quartz and other materials suggests a lack of consideration for the practicalities While the fabrication of such materials may be possible, the requirement for advanced manufacturing equipment (technologies of a complexity far greater than that of the sensors being discussed) and the feasibility of such processes in a lunar environment are not adequately addressed.
RESPONSE: The reviewer did not suggest what “advanced manufacturing equipment…[with]…complexity far greater than that of the sensors” that they were thinking of. Lines 372-376 describe the requirements for the production of quartz – it requires a sealed pressure vessel but modest temperatures with pressure and temperatures sensors. This is not more complex than a PMT.
In general, I have clearly defined the scope of the article in several places:
Line 217 clearly states “As we focus on material availability….investigation includes feasible chemical processing pathways….”
Line 227 clearly states “roadmap to manufacturing….emphasis on lunar materials and aspects of their chemical processing rather than manufacturing techniques….”
Line 231 states “We are examining resource restrictions aspect of manufacturing sensors….” Line 582 states explicitly that manufacturing of PMTs is not within scope of the article.
Reviewer Reply: I think that it is disingenuous to present chemical processing pathways in a vacuum without any consideration of the actual manufacturing processes required to accomplish it. Looking to the chemistry alone makes in-situ manufacturing look easy, when the reality is that the technical challenges associated with the chemical processing are vast. Furthermore, the processes the author suggests in text, such as HCl leaching of anorthosite, are chemically possible, however the challenges with implementing such techniques are non-trivial. What happens if, during a design trade off study, an electrochemical approach is employed instead of a hydrometallurgical one?
RESPONSE 2: I fail to see how the paper is disingenuous – at multiple points, I have clearly defined the scope of the paper. My goal is to show that there is a pathway to complex manufacturing of sensors by examining the basic materials constraints on the Moon. I am not aware of anyone examining this problem before. My intent is not to present a fait accompli but a plausible pathway. However, to ensure that your concerns regarding triviality are addressed, in the conclusion, I have added a clear and explicit statement thus:
“The methods we address and propose are not fait accompli – as in all things, the devil will be in the details. To be sure, showing the chemical processes does not address the engineering implementations. For example, the step from laboratory demonstration to practical realisation in a lunar payload is a significant one. There are multiple considerations yet to be addressed including (but not exclusively): (i) feedback control of chemical processing which is inherently nonlinear and time-varying; (ii) material processing, handling and throughput plumbing between processing stations must accommodate transport of different forms of product and feedstock; (iii) automation of transport and throughputs of samples robotically including fault handling; (iv) manufacturing planning, allocation and monitoring for part fabrication; (v) at all stages, analytical instrument integration for monitoring of processing conditions and product integrity.”
Regarding our processing choices, we have already traded down on an efficient lunar industrial ecology to minimise reagents, waste, energy, etc (in the Appendix). We have performed HCl leaching of highland simulant in the lab followed by molten salt electrolysis. These are referenced and details of challenges are described therein. To be sure, it is possible that other methods may be found to be superior – I think the reviewer is referring to direct regolith decomposition through molten regolith electrolysis. Other methods such as this are discussed in some of the references but our default scenario is the lunar industrial ecology outlined. I have argued elsewhere that this is currently the most efficient means to support a lunar infrastructure – but that is not a discussion for this paper which is focused on bio-inspired sensors.
[Reviewer Reply]: At this stage, literally any manufacturing activity on the lunar surface will be extremely complicated. The production of the raw materials needed to manufacture the sensors discussed in this work has never been demonstrated anywhere in the solar system, aside from on the surface of the Earth. The current drive in the ISRU community is to produce water or oxygen from resources, which is proving difficult enough as is. The production of high-purity metals and/or Si-based plastics is another massive technical challenge. The author has provided discussion of how these processes might work, however understanding the chemistry and actually implementing such processes on the Moon are vastly different. Stating that quartz can be produced using “A sealed pressure vessel but with modest temperatures with pressure and temperature sensors” may be technically true, but greatly oversimplifies the engineering challenges associated with it. I have no doubt that in-situ manufacturing on the Moon will one day be a reality, but broad oversimplifications of the technologies needed to get to that stage are unhelpful, especially when presenting a new area of science and engineering to the Biomimetics community that may otherwise be unfamiliar with the intricacies and challenges of ISRU.
RESPONSE 2: I understand the reviewer’s concern but my impression is that the reviewer is suggesting that since we haven’t successfully demonstrated ISRU (except MOXIE), we should not bother to figure out anything more complex. I do not agree – I think we can and should define a long-term goal (lunar industrialisation) and figure out a means (beginning with carving out a pathway with more detail than a roadmap) to achieve it. I have presented a pathway to one aspect of it. I am fully aware of the complexities of chemical processing as we have performed some of these processes in the lab. And we haven’t tried it on the Moon but we have shown that the basic processes of Al extraction work. This gives us some confidence that it will be possible – that is our job as engineers to solve problems as we progress (indeed, we are scoping to build a lunar payload but that is not the subject of this paper). I have included references that do go into some detail on intricacies for readers to explore if they so wish. Delving deep into details of ISRU is not the focus here – bio-inspiration is. However, you are correct that I should mention some of the difficulties that arise – to that end, I have added a short set of statements on some of these aspects in the conclusion that is copied above. That clearly states that there are many issues to be resolved….